# Mechanism of ribosome rescue by ArfA and RF2

**Gabriel Demo[1], Egor Svidritskiy[1], Rohini Madireddy[2†], Ruben Diaz-Avalos[3], Timothy Grant[3], Nikolaus Grigorieff[3], Duncan Sousa[4], Andrei A Korostelev[1,2]\***

[1]RNA Therapeutics Institute, University of Massachusetts Medical School, Worcester, United States; [2]Department of Biochemistry and Molecular Pharmacology, University of Massachusetts Medical School, Worcester, United States; [3]Janelia Research Campus, Howard Hughes Medical Institute, Ashburn, United States; [4]Department of Biological Science, Florida State University, Tallahassee, United States

**Abstract** ArfA rescues ribosomes stalled on truncated mRNAs by recruiting release factor RF2, which normally binds stop codons to catalyze peptide release. We report two 3.2 Å resolution cryo-EM structures – determined from a single sample – of the 70S ribosome with ArfA•RF2 in the A site. In both states, the ArfA C-terminus occupies the mRNA tunnel downstream of the A site. One state contains a compact inactive RF2 conformation. Ordering of the ArfA N-terminus in the second state rearranges RF2 into an extended conformation that docks the catalytic GGQ motif into the peptidyl-transferase center. Our work thus reveals the structural dynamics of ribosome rescue. The structures demonstrate how ArfA 'senses' the vacant mRNA tunnel and activates RF2 to mediate peptide release without a stop codon, allowing stalled ribosomes to be recycled.

**\*For correspondence:** andrei. korostelev@umassmed.edu

**Present address:** [†]Medicago, Durham, United States

## Introduction

A translating ribosome stalls when it encounters the end of a non-stop mRNA, truncated during cellular stress or homeostasis, by premature transcription termination or mRNA cleavage or other mechanisms (*Hayes and Keiler, 2010*; *Keiler, 2015*). The stalled ribosome contains peptidyl-tRNA in the P site, whereas the aminoacyl-tRNA (A) site is unoccupied (*Ito et al., 2011*). Bacteria have evolved ribosome-rescue pathways to release the nascent peptide and re-enable the stalled ribosome for translation (for a review, see ref. [*Keiler, 2015*]). The ArfA (alternative rescue factor A) pathway is essential in trans-translation-deficient cells (*Chadani et al., 2010*) and is thought to function as a backup mechanism for trans-translation in enterobacteria (*Garza-Sánchez et al., 2011*; *Chadani et al., 2011*; *Schaub et al., 2012*). ArfA is a small protein (~70 aa in most organisms), with only 47 amino acids sufficient for function, as shown for *E. coli* ArfA truncations (*Garza-Sánchez et al., 2011*). ArfA recruits release factor RF2 to rescue stalled ribosomes (*Chadani et al., 2012*; *Shimizu, 2012*). RF2 normally mediates translation termination at UGA or UAA stop codons by binding the stop codon and catalyzing peptidyl-tRNA hydrolysis and release of the nascent peptide (*Craigen and Caskey, 1987*; *Korostelev, 2011*). RF2 has remarkable specificity toward stop codons (*Freistroffer et al., 2000*) and does not function alone on truncated mRNA (*Chadani et al., 2012*; *Shimizu, 2012*). In this work, we asked how ArfA and RF2 sense the stalled ribosome, and how ArfA aids RF2 to catalyze peptide release in the absence of a stop codon.

To better understand ribosome rescue by ArfA and RF2, we formed an *E. coli* 70S ribosome rescue complex with mRNA truncated after an AUG codon in the P site, tRNA^fMet, ArfA and RF2, and captured images of complexes by electron cryo-microscopy (cryo-EM; see Materials and methods). Unsupervised classification of a single cryo-EM dataset (*Figure 1—figure supplement 1*) using

FREALIGN (*Grigorieff, 2016*) revealed two ribosome structures with both ArfA and RF2 bound in the A site (Structures I and II; *Figure 1* and *Figure 1—figure supplement 2* and *Table 1*). Structure I contains a compact RF2 (*Figure 1A and C*) and is represented by ~26% of ribosome particles in the cryo-EM dataset. Structure II contains an extended RF2 (*Figure 1B and D*; ~18% of ribosomes). Both structures contain tRNAs in the P and E (exit) sites and adopt a non-rotated conformation (*Yusupov et al., 2001*; *Korostelev et al., 2006*; *Selmer et al., 2006*), similar to that in translation termination complexes (*Korostelev et al., 2008*; *Laurberg et al., 2008*; *Weixlbaumer et al., 2008*). High-resolution maps (*Figure 1E and F* and *Figure 1—figure supplements 3–5*) allowed de novo modeling of ArfA (*Figure 1E* and *Figure 1—figure supplement 3*) and detailed structure determination of RF2 (*Figure 1E and F* and *Figure 1—figure supplement 4*) in each ribosome structure. The molecular interactions and conformational rearrangements inferred from Structures I and II provide the structural basis for ArfA-mediated ribosome rescue, as described below.

## Results and discussion

### ArfA C-terminus occupies the mRNA tunnel to sense the stalled ribosome

Sequence alignment of several hundreds of bacterial ArfA homologs reveals conserved hydrophobic N-terminal and positively charged C-terminal regions (*Figure 2*). The high conservation of these regions implicates their functions in protein and RNA interactions, respectively. In Structures I and II, the conformations of the N-terminal region differ as described in the following sections, but the rest of ArfA is similar. The mid-region of ArfA (His21 to Glu30; *E. coli* numbering is used) lies in the A site (*Figure 1C and D*). ArfA leaves a ~12 Å gap in the codon-binding region, sufficient to accommodate one or two nucleotides of mRNA following the P-site codon but not a longer mRNA, consistent with the reduced efficiency of ArfA-mediated release on mRNAs that extend three or more nucleotides beyond the P site (*Shimizu, 2012*; *Zeng and Jin, 2016*).

The C-terminal region of ArfA occupies the mRNA tunnel between the head and body of the 30S subunit (*Figure 2—figure supplements 1A* and *2*), as suggested by previous hydroxyl-radical probing studies (*Kurita et al., 2014*). The mRNA tunnel is primarily formed by the negatively charged 16S rRNA backbone. ArfA is stabilized by electrostatic interactions, as positively charged amino acids comprise nearly half of the ArfA residues within the mRNA tunnel (aa 31–48; *Figure 2B*). ArfA residues Lys44 to Arg48 approach the mRNA tunnel entry at the solvent side of the 30S subunit, formed by 16S rRNA and proteins S3, S4 and S5 (*Figure 2* and *Figure 2—figure supplement 1A*). Consistent with ArfA structure prediction (*Kim et al., 2004*; *Yang et al., 2015*), the tail of ArfA appears to form a short α-helix (aa ~50–55) next to S5; however, the resolution of the map is not sufficient to build an unambiguous structural model (*Figure 2—figure supplement 2*). This suggests conformational disorder of the C-terminus at the entrance to the mRNA tunnel. Our structures are consistent with biochemical studies (*Chadani et al., 2011*), which showed that ArfA is functional with a C-terminal truncation at Asn47, but further shortening inactivates ArfA. In particular, truncations following Met40—removing at least five basic amino acids that bind in the tunnel—abrogate ArfA-mediated release by reducing ArfA affinity for the 70S ribosome (*Chadani et al., 2011*).

We compared ArfA with proteins ArfB (also known as YaeJ; [*Gagnon et al., 2012*]) and SmpB (*Neubauer et al., 2012*; *Ramrath et al., 2012*), which mediate alternative rescue pathways (reviewed in [*Keiler, 2015*]). In ArfB-mediated release and SmpB-mediated trans-translation, the proteins sense the stalled ribosomes by occupying the mRNA tunnel (*Gagnon et al., 2012*; *Neubauer et al., 2012*). Consistent with sequence divergence among the C-termini of ArfA, ArfB, and SmpB, however, each interacts with the mRNA tunnel differently. ArfB performs peptide release when its C-terminus forms a long α-helix in the tunnel (*Gagnon et al., 2012*). SmpB bound with tmRNA in preparation for trans-translation, forms a shorter α-helix, well-resolved near the mRNA entry at the 30S solvent surface (*Neubauer et al., 2012*). SmpB is similar to ArfA in that both proteins are sensitive to the mRNA occupancy of the A site (*Ivanova et al., 2004*; *Shimizu, 2012*; *Zeng and Jin, 2016*), whereas ArfB can function on stalled ribosomes with either the vacant or occupied A site (*Handa et al., 2011*). Diverged sequences, structures and binding modes of these proteins likely reflect distinct affinities and sensitivities to stress conditions, in keeping with their roles in mediating distinct ribosome rescue pathways (*Figure 2—figure supplement 1*).

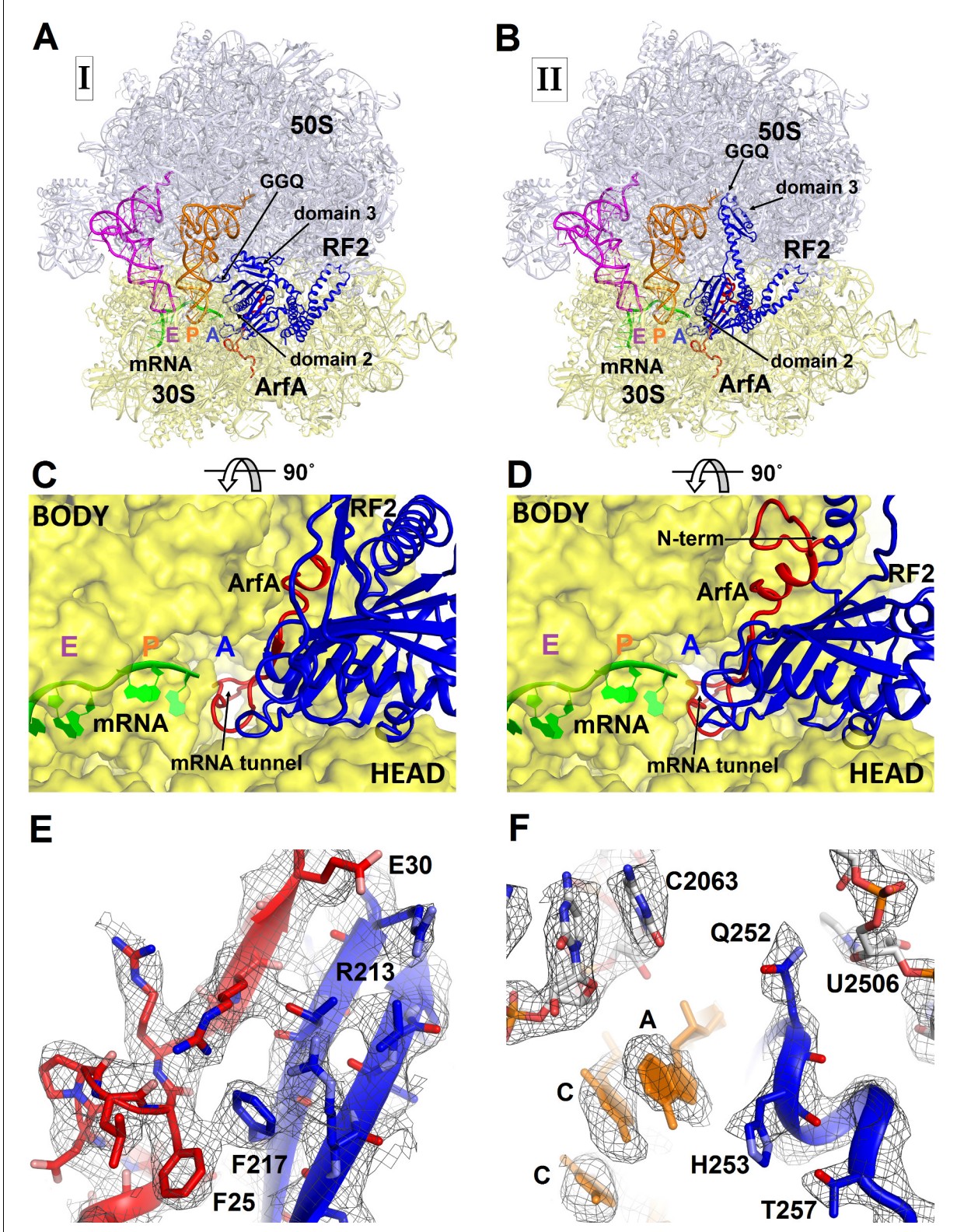

**Figure 1.** 3.2 Å resolution cryo-EM structures of *E. coli* 70S ribosome bound with ArfA and release factor RF2. (A) Structure I with RF2 in a compact conformation; (B) Structure II with RF2 in an extended conformation. Domains 2 and 3 and the GGQ motif of RF2 are labeled. (C) and (D) A close-up view down the mRNA tunnel, showing RF2 and ArfA in the A site of Structure I (C) and Structure II (D). The body and head domains of the 30S subunit are labeled. (E) Extended β-sheet formed by ArfA (red model) and RF2 (blue model). Cryo-EM map (gray mesh) is shown for Structure II at σ = 2.5. (F)

*Figure 1 continued on next page*

*Figure 1 continued*

Peptidyl-transferase center bound with the $^{250}$GGQ$^{252}$ motif of RF2 in Structure II. Cryo-EM map (gray mesh) is shown at σ = 2.5 for RF2 and at σ = 4.5 for 23S ribosomal RNA and the $^{74}$CCA$^{76}$ end of the P-site tRNA. The maps were sharpened by applying the B-factor of −120 Å$^2$. Additional views of cryo-EM density are available in *Figure 1—figure supplements 1–5*. In all panels, the large 50S ribosomal subunit is shown in gray/light-blue; the small 30S subunit in yellow; mRNA in green; E-site tRNA in magenta; P-site tRNA in orange; ArfA in red and RF2 in blue.

The following figure supplements are available for figure 1:

**Figure supplement 1.** Schematic of cryo-EM refinement and classification.

**Figure supplement 2.** Cryo-EM densities of Structures I and II.

**Figure supplement 3.** Cryo-EM densities corresponding to N- and C-terminal regions of ArfA in Structures I and II.

**Figure supplement 4.** Cryo-EM densities corresponding to functional regions of RF2 in Structures I and II.

**Figure supplement 5.** Cryo-EM densities of ribosomal RNA in Structure II.

## ArfA N-terminus is disordered in the presence of a compact (inactive) RF2 conformation

In Structure I, only the central and C-terminal parts of ArfA are visible, indicating that the N-terminal region (aa 2–16) is disordered. RF2 adopts a compact conformation (*Figure 3A*) similar to that of free (ribosome-unbound) RF2 observed in crystal structures (*Figure 3B*) (*Vestergaard et al., 2001*; *Zoldák et al., 2007*). By contrast, in canonical termination complexes formed on stop codons, release factors have only been observed in an extended (open) conformation (*Korostelev et al., 2008*; *Weixlbaumer et al., 2008*). During translation termination, codon-recognition determinants in domain 2 (including the conserved $^{205}$SPF$^{207}$ motif) of RF2 bind the stop codon in the A site of the 30S subunit. Helix α7 of domain 3 bridges the ribosomal subunits, placing the catalytic GGQ motif of domain 3 within the peptidyl-transferase center of the 50S subunit (*Korostelev et al., 2008*; *Weixlbaumer et al., 2008*). In the ArfA-bound Structure I, however, helix α7 packs on the β-sheet of domain 2 near the 30S subunit (*Figure 3A and B*). In this compact conformation, the loop that contains the $^{250}$GGQ$^{252}$ motif of RF2 lies to the side of the β-sheet (near aa 165–168) of domain 2, facing the anticodon-stem loop and the D stem of the P-site tRNA. As such, the GGQ motif is roughly 70 Å away from its catalytically engaged position within the peptidyl-transferase center (*Figure 3C and D*). Poor resolution of the catalytic GGQ residues (*Figure 1—figure supplement 4A*) at the tip of the loop suggests local structural flexibility, similar to that seen in crystal structures of free release factors (*Vestergaard et al., 2001*; *Shin et al., 2004*; *Zoldák et al., 2007*).

The codon-recognition domain 2 of RF2 is positioned differently from that in canonical termination complexes. The domain is withdrawn from the A site, such that the SPF motif and other codon-recognition residues lie ~5 Å away from their positions in termination complexes bound to a stop codon (*Figure 4* and *Figure 1—figure supplement 4A*). This position results from the mid-region of ArfA being sandwiched between domain 2 of RF2 and the decoding center. Here, the backbone of ArfA residues 25–29 binds to RF2 residues 213–217 within the β-sheet of domain 2 (*Figure 1E*), forming an extended β-sheet platform (*Figure 4A and D*). The conformation of the decoding center in Structure I differs from that in canonical termination complexes. In 70S termination complexes, the decoding center interacts with the switch loop of RF2 (aa 315–323), which bridges the codon-recognition and catalytic domains (*Korostelev et al., 2008*; *Weixlbaumer et al., 2008*). A1492 bulges from helix 44 (h44) of 16S rRNA and stacks on conserved Trp319 of the switch loop (*Figure 4C*), stabilizing the extended conformation of RF2 on the ribosome (*Figure 4F*) (*Korostelev et al., 2008*; *Weixlbaumer et al., 2008*). In Structure I, however, A1492 and A1493 lie inside h44 and are sandwiched between Pro23 of ArfA and A1913 of helix 69 of 23S rRNA (*Figure 4A*). The RF2 switch loop is poorly ordered. Putative density suggests that Trp319 is placed >10 Å away from its position in the termination complex and does not contact the decoding-center nucleotides (*Figure 4D* and *Figure 1—figure supplement 4A*).

**Table 1.** Cryo-EM data collection and refinement statistics.

| | Structure I | Structure II |
|---|---|---|
| PDB code | 5U9G | 5U9F |
| EMDB code | EMD-8522 | EMD-8521 |
| **Data collection** | | |
| EM equipment | FEI Titan Krios | FEI Titan Krios |
| Voltage (kV) | 300 | 300 |
| Detector | DE-20 | DE-20 |
| Pixel size (Å) | 1.215 | 1.215 |
| Electron dose ($e^-$/Å$^2$) | 61 (used 30) | 61 (used 30) |
| Defocus range (μm) | −0.5 to −3.0 | −0.5 to −3.0 |
| **Reconstruction** | | |
| Software | Frealign v9.10–9.11 | Frealign v9.10–9.11 |
| Number of particles used | 139,861 | 96,070 |
| Final resolution (Å) | 3.15 | 3.15 |
| Map-sharpening $B$ factor (Å$^2$) | −92.1 | −89.9 |
| **Model building** | | |
| Software | Coot | Coot |
| **Model composition** | | |
| Non-hydrogen atoms | 152671 | 152841 |
| Protein residues | 6561 | 6576 |
| RNA bases | 4729 | 4729 |
| Ligands ($Zn^{2+}$/$Mg^{2+}$) | 1/348 | 1/360 |
| **Refinement** | | |
| Software | RSRef and Phenix | RSRef and Phenix |
| Correlation Coeff (%; Phenix) | 82.90 | 80.40 |
| R-factor (RSRef) | 0.187 | 0.194 |
| **Validation (proteins)** | | |
| MolProbity score | 2.3 | 2.2 |
| Clash score, all atoms | 12.6 | 13.0 |
| Good rotamers (%) | 93.7 | 94.4 |
| Ramachandran-plot statistics (%) | | |
| Favored (overall) | 88.4 | 88.2 |
| Allowed (overall) | 10.6 | 10.8 |
| Outlier (overall) | 1.0 | 1.0 |
| Favored (ArfA) | 86.7 | 88.9 |
| Allowed (ArfA) | 13.3 | 11.1 |
| Outlier (ArfA) | - | - |
| Favored (RF2) | 86.4 | 92.5 |
| Allowed (RF2) | 13.6 | 7.2 |
| Outlier (RF2) | - | 0.3 |
| R.m.s. deviations | | |
| Bond length (Å) | 0.006 | 0.005 |
| Bond angle (°) | 0.852 | 0.864 |

*Table 1 continued on next page*

*Table 1 continued*

|  | Structure I | Structure II |
|---|---|---|
| Validation (RNA) |  |  |
| Correct sugar puckers (%) | 99.9 | 99.9 |
| Good backbone conformation (%) | 85.2 | 85.3 |

Taken together, Structure I describes a 70S rescue complex in which ArfA stabilizes a compact form of RF2 resembling that of free RF2. The remote position of RF2's catalytic GGQ motif—away from the peptidyl-transferase center—indicates that Structure I represents an inactive form of the 70S rescue complex.

## The ordering of the ArfA N-terminus is coupled with an extended (active) RF2 conformation

Structure II features an extended conformation of RF2 (*Figures 1B*, *3C and D*), stabilized by interactions with the ordered N-terminal region of ArfA (*Figures 2*, *4B and E*). The N-terminus of ArfA forms a minidomain, which packs between helix 69 of 23S rRNA, the $\beta$-sheet of domain 2, and

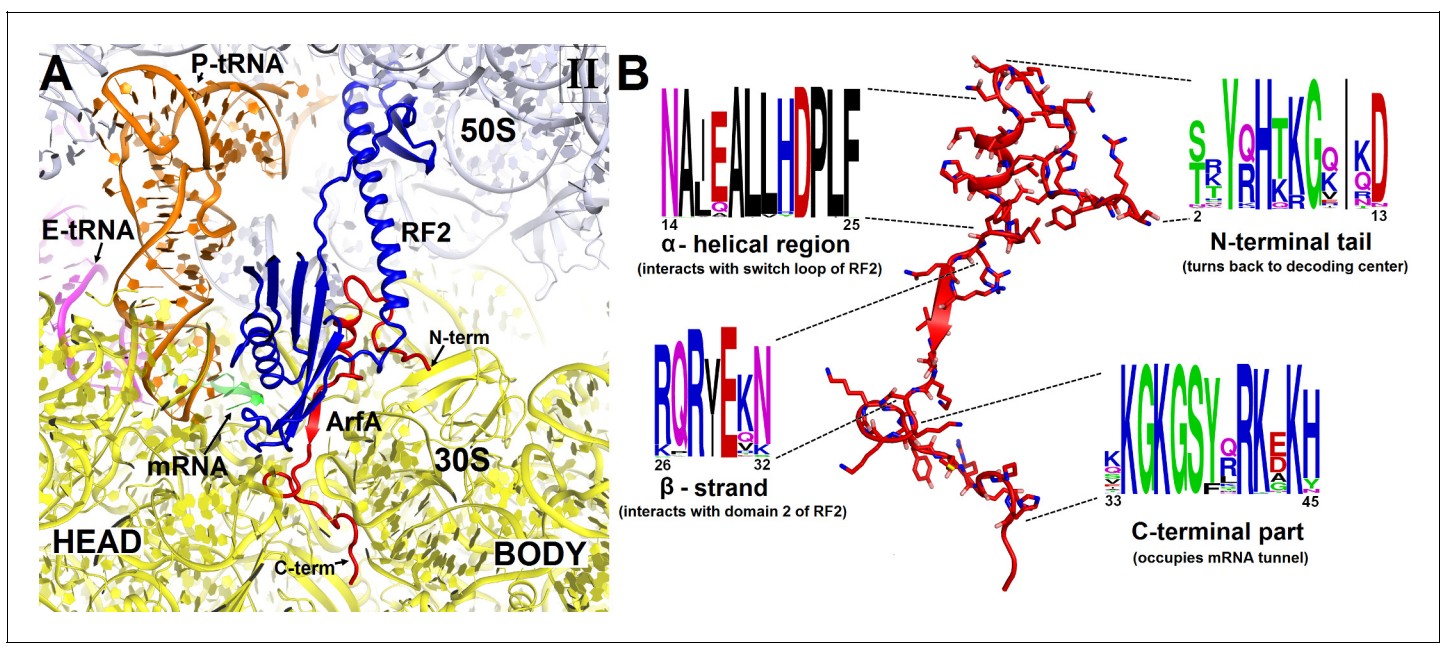

**Figure 2.** Structure and sequence of ArfA. (A) Close-up view of the intersubunit space and mRNA tunnel occupied by ArfA in Structure II. The color-coding is the same as in *Figure 1*. The head and body domains of the 30S subunit are labeled. (B) Sequence and structure of ArfA, shown in the same orientation as in (A). Sequence conservation among ~400 non-redundant bacterial ArfA homologs is shown for four ArfA regions (see Materials and methods). The color code for amino acid type in the sequence logo is the following: green – polar, purple – neutral, blue – basic, red – acidic and black – hydrophobic residues.

The following figure supplements are available for figure 2:

**Figure supplement 1.** Comparison of mRNA tunnel occupancies by ribosome rescue proteins ArfA, ArfB and SmpB (shown in red).

**Figure supplement 2.** Putative α-helical structure of the C-terminal region of ArfA at the entry of the mRNA tunnel (near protein S5) in Structures I and II (Structure II is shown).

**Figure supplement 3.** Structure of the N-terminal tail of ArfA near the decoding center in Structure II.

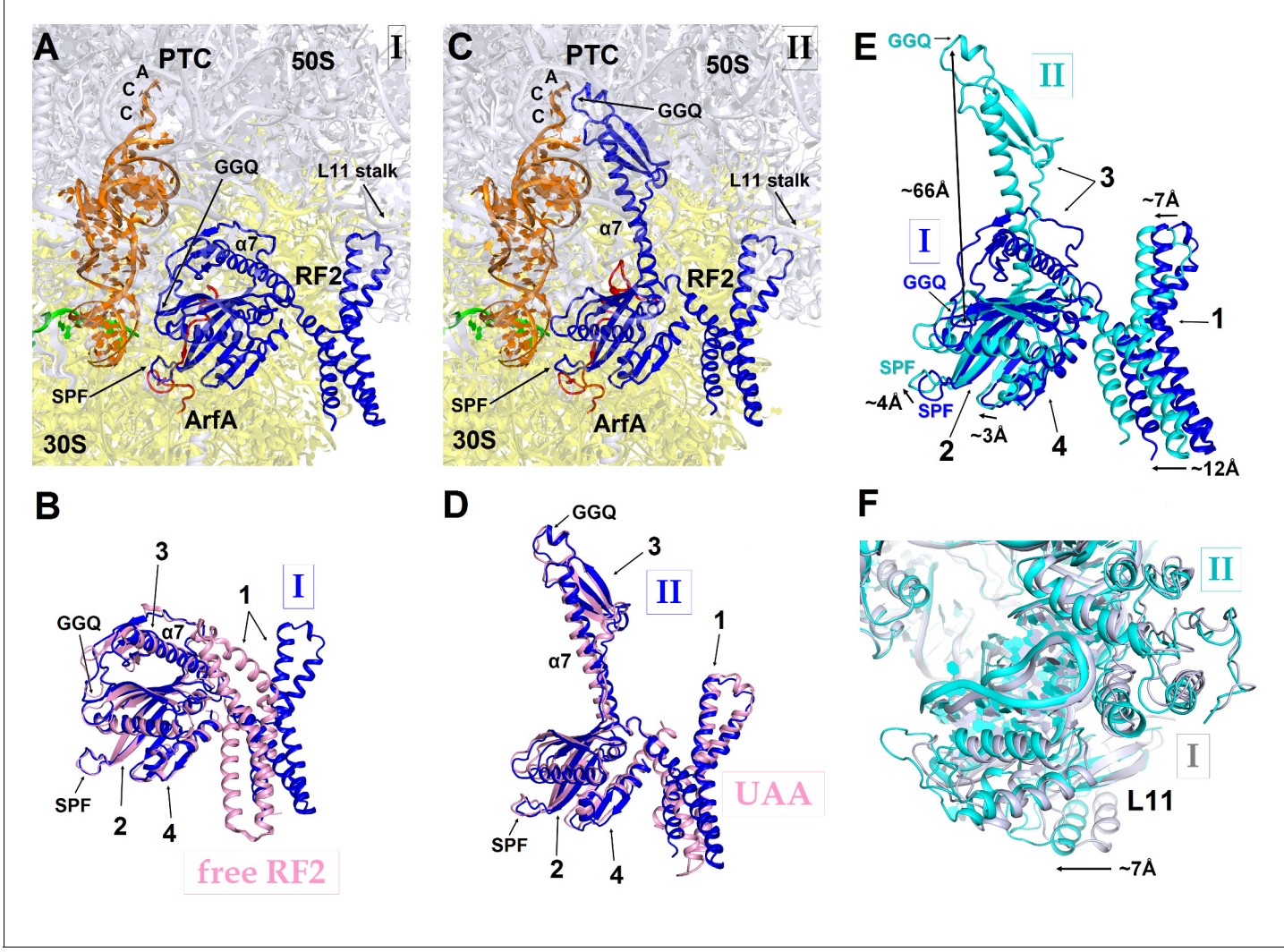

**Figure 3.** RF2 adopts two distinct conformations in Structures I and II. (A) The P and A sites of Structure I. ArfA is shown in red; RF2 in blue; mRNA in green; P-tRNA in orange; 30S subunit in yellow; and 50S subunit in light blue. (B) Superposition of RF2 from Structure I (blue) with the crystal structure of free (ribosome-unbound) *E. coli* RF2 (PDB 1GQE) (pink). Relative positions of the codon-recognition superdomain (domains 2 and 4) and catalytic domain 3 are nearly identical. The positions of domain 1 differ; this domain in both Structures I and II interacts with the L11 stalk at the 50S subunit shown in panels (A), (C) and (F). (C) The P and A sites of Structure II. The color coding is as in panel (A). (D) Superposition of extended RF2 in Structure II (blue) with *Thermus thermophilus* RF2 in the canonical termination complex formed on the UAA stop codon (PDB 4V67) (pink). The superposition was performed by structural alignment of 16S ribosomal RNAs. RF2 adopts similar conformations but domains 2 and 3 are positioned slightly differently with respect to the 30S subunit in the rescue complex II and in the termination complex (see also *Figure 4*). (E) Superposition of RF2 in Structures I (blue) and II (cyan), achieved by structural alignment of the 16S ribosomal RNAs. Conformations of RF2 and positions relative to the 30S subunit differ between Structures I and II, as RF2 in Structure II binds deeper in the A site; differences in positions of RF2 regions are labeled with arrows. (F) Different positions of the L11 stalk, which interacts with domain 1 of RF2, in Structures I (light blue) and II (cyan), suggesting movement of the stalk together with domain 1 (E) upon RF2 activation. The view is similar to that shown in panels A, C and E. In panels (B), (D) and (E), the Arabic numerals label the domains of RF2.

extended helix α7 of RF2 (*Figure 2—figure supplement 3*). The tip of the ArfA minidomain (at aa 12–13) protrudes from the decoding center toward the 50S subunit, whereas the N-terminus folds back toward the decoding center, consistent with recent hydroxyl-radical probing studies (*Kurita et al., 2014*). Here, Thr7 and Lys8 bind h44 (at A1410) of 16S rRNA and the N-terminal amino acids bind the β-sheet of S12 (*Figure 2—figure supplement 3*).

A hydrophobic patch in the N-terminal minidomain of ArfA (*Figure 2B*) — formed by Leu19, Leu24, and Phe25—binds RF2 at Trp319 of the rearranged switch loop (*Figure 4B and E* and

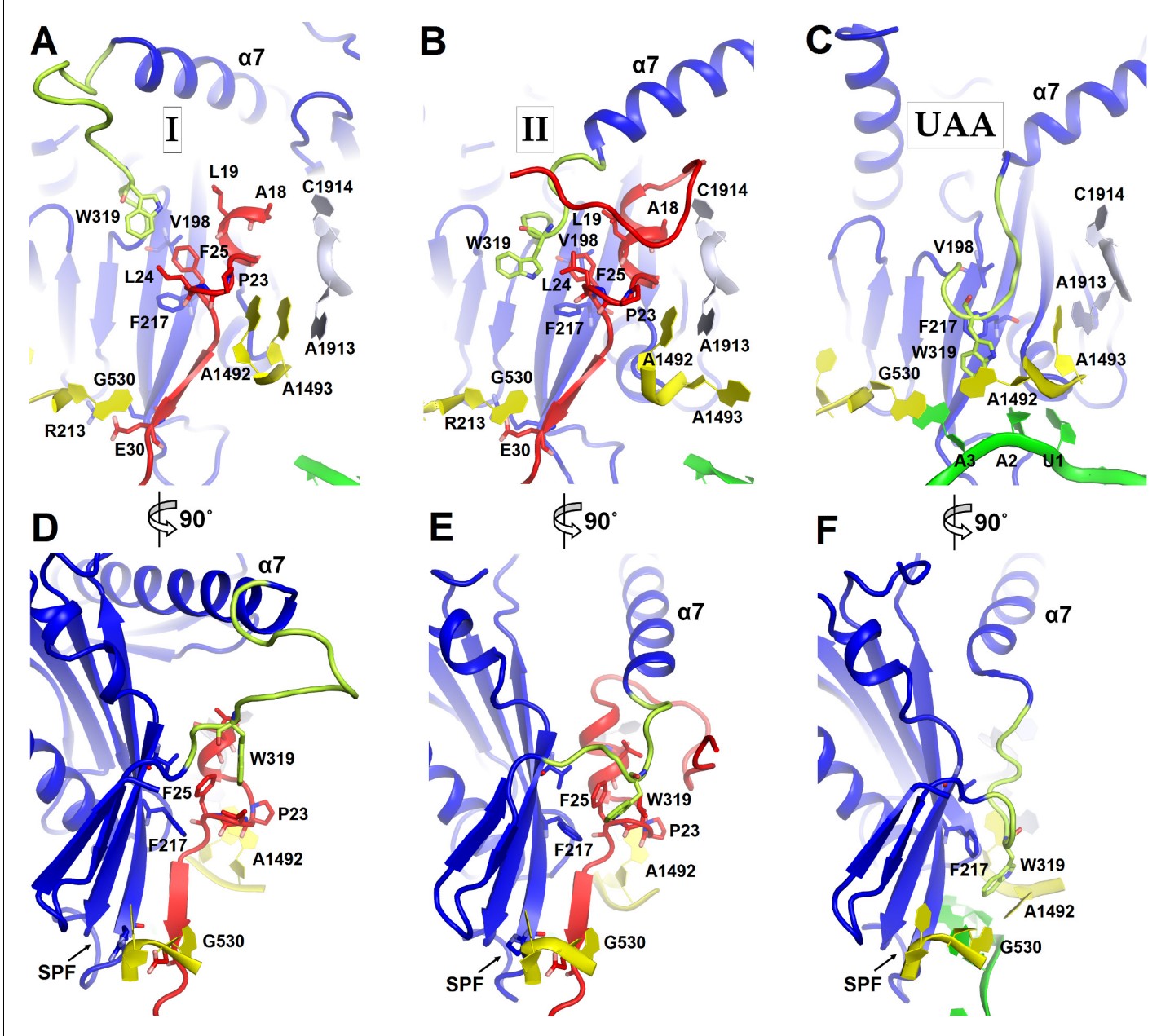

**Figure 4.** Positions of the codon-recognition domain (blue) and switch loop (yellow-green) of RF2 in Structures I and II (this work) and in the translation termination complex formed on the UAA stop codon (*Korostelev et al., 2008*). (A–C) Detailed view of the decoding center of Structure I (A), Structure II (B), and canonical termination complex formed on the UAA stop codon (C). (D–F) 90-degree rotated views (relative to those shown in panels A–C) of the decoding center in Structure I (D), Structure II (E) and the UAA-containing termination complex (F). The switch loop of RF2, which carries the conserved Trp319, adopts different positions in these three structures. Key structural features and residues of ArfA, RF2, mRNA stop codon and the ribosomal decoding center are labeled. ArfA is shown in red; RF2 in blue (RF2 switch loop in yellow-green); mRNA in green; 30S nucleotides in yellow; and 50S nucleotides in light blue.

*Figure 1—figure supplement 4B*). These interactions explain the strict dependence of ArfA on RF2, rather than on the second release factor RF1 (*Chadani et al., 2012*; *Shimizu, 2012*), whose switch loop is diverged from that of RF2 and lacks tryptophan (*Korostelev et al., 2010*). The hydrophobic patch in the N-terminal minidomain also binds RF2 at Val198 and Phe217 of the β-sheet of the codon-recognition domain 2 (*Figure 4B and E* and *Figure 1—figure supplement 4B*). In this

configuration, the codon-recognition domain of RF2 partially settles into the decoding center, but remains ~3 Å from its position in canonical termination complexes bound to a stop codon. As in Structure I, the SPF motif remains unbound to the ribosome or ArfA. This observation explains why mutation of SPF motif residues—critical for stop-codon recognition—do not disrupt ArfA-mediated peptide release (*Chadani et al., 2012*).

The position of the ArfA N-terminal minidomain between RF2's domain 2 and helix α7 of domain 3 results in docking of domain 3 into the peptidyl-transferase center (*Figures 1*, *3C and D*). The opening of domain 3 is accompanied by movement of domain 1 of RF2 (*Figure 3E*), which binds the L11 stalk and shifts the L11 stalk by ~7 Å toward the A site, relative to that in Structure I (*Figure 3F*). The coordinated shift of L11 and domain 1 likely contribute to the movement of domain 3 toward the PTC, echoing the involvement of the L11 stalk in canonical termination by RF2 (*Xu et al., 2002*; *Sato et al., 2006*). The catalytic GGQ motif of domain 3 binds in the peptidyl-transferase center with the catalytic backbone amide of Gln252 (*Korostelev et al., 2008*; *Santos et al., 2013*) proximal to the ribose of the terminal A76 of P-site tRNA (*Figure 1F*, *Figure 1—figure supplement 4B*). The conformation of the peptidyl-transferase center is nearly identical to that seen in canonical translation termination complexes (*Korostelev et al., 2008*; *Weixlbaumer et al., 2008*). Structure II therefore represents an activated state of the ArfA•RF2-bound ribosome rescue complex.

Structures I and II are in agreement with previous biochemical and mutagenesis studies, as we have described above (*Chadani et al., 2010*, *2012*; *Shimizu, 2012*; *Kurita et al., 2014*; *Zeng and Jin, 2016*). An ArfA-inactivating mutation has been identified (*Chadani et al., 2010*). Mutation of Ala18 of the N-terminal ArfA minidomain to threonine prevents ArfA-mediated peptide release without disrupting RF2 binding (*Shimizu, 2012*). In Structure II, Ala18 lies in the hydrophobic core of the N-terminal fold, tightly packed between the nucleobase of C1914 of h69 and Ile11 of ArfA (*Figure 2—figure supplement 3*). The substitution to the larger threonine residue is likely incompatible with the ordered N-terminal fold of ArfA and the extended conformation of RF2. The mutation should, however, be compatible with an inactive ribosome rescue complex (Structure I), consistent with RF2 binding and the lack of catalytic activity.

## Structural mechanism of ribosome rescue by ArfA and RF2

During ribosome rescue, the release of the nascent peptide should strictly coordinate with the recognition of a vacant mRNA tunnel. Our cryo-EM analysis indicates that ribosomes formed on a truncated mRNA in the presence of ArfA and RF2 adopt at least three states, including Structures I, II and the ribosome with a vacant A site (*Figure 1—figure supplement 1*). These states suggest a structure-based model for stepwise release of nascent peptides from stalled ribosomes during ArfA-mediated ribosome rescue (*Figure 5*), summarized as an animation (*Video 1*, also available at http://labs.umassmed.edu/korostelevlab/movarfa.gif).

In the absence of an A-site codon, peptidyl-tRNA-bound ribosomes cannot bind aminoacyl-tRNA (*Figure 5A*). They are recognized by ArfA, which binds the empty mRNA tunnel and recruits a compact (inactive) conformation of RF2 (*Figure 5B*). Biochemical studies showed that ArfA can bind the ribosome without RF2 (*Kurita et al., 2014*), suggesting that ArfA precedes RF2. We do not observe ribosomes bound with ArfA alone. This indicates that ArfA has a substantially higher affinity in the presence of RF2 and predominantly binds as a complex with RF2. Additional biochemical studies are required to elucidate this step. The initial rescue complex (Structure I) then samples an extended (active) conformation of RF2 coupled to ordering of the N-terminal minidomain of ArfA at the ribosomal decoding center (Structure II; *Figure 5C*). Although such structural rearrangement of release factors RF1 and RF2 has not been directly observed during canonical termination, it has been suggested based on structural comparisons (*Rawat et al., 2003*; *Laurberg et al., 2008*) and biochemical studies that revealed dynamic behaviors of domain 3 and the switch loop (*He and Green 2010*; *Trappl and Joseph, 2016*). A low-resolution small-angle X-ray scattering study proposed a predominant extended conformation of free *E. coli* RF1 (*Vestergaard et al., 2005*). However, subsequent characterization of *E. coli* and *Thermus thermophilus* RF2 (*Zoldák et al., 2007*) and *E. coli* RF1 (*Trappl and Joseph, 2016*) demonstrated predominance of the closed form in solution, consistent with crystal structures of RF2 and RF1 (*Vestergaard et al., 2001*; *Shin et al., 2004*; *Zoldák et al., 2007*). The tmFRET study that followed changes in inter-domain distances in RF1, found that the pre-termination ribosome initially interacts with a closed form of RF1 and induces a large-scale

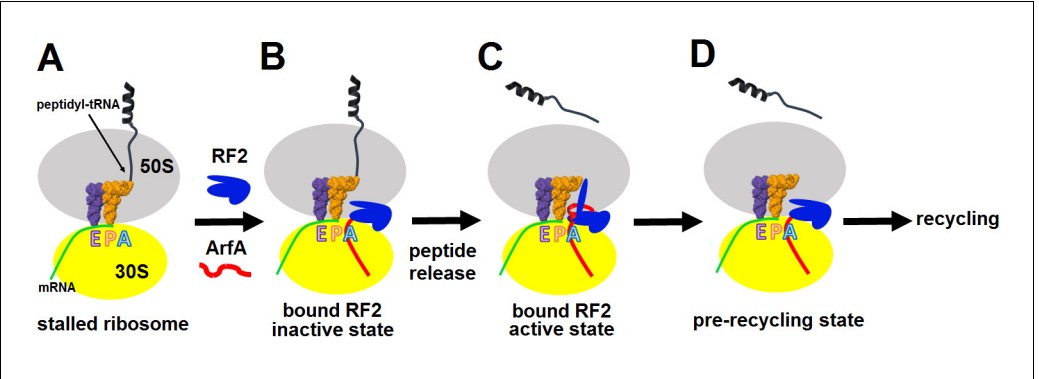

**Figure 5.** Mechanism of ArfA-mediated rescue of ribosomes stalled on truncated mRNA. (A–B). ArfA senses the stalled ribosome by binding its C-terminal portion in the vacant mRNA tunnel, recruiting RF2 in an inactive conformation (as in Structure I). (C) Folding of the N-terminal minidomain of ArfA is coupled with the opening of RF2, placing the GGQ motif into the peptidyl-transferase center (as in Structure II) and catalyzing peptidyl-tRNA hydrolysis. (D) Following peptide release, ArfA and RF2 (likely in a compact conformation) depart, preparing the ribosome for recycling.

rearrangement (*Trappl and Joseph, 2016*). The opening of a release factor is therefore a plausible conserved mechanism to separate the 'decoding' and 'catalytic' functions of release factors.

In the 70S•ArfA•RF2 complex, the opening of RF2 places the GGQ motif into the peptidyl-transferase center (Structure II), where it catalyzes peptidyl-tRNA hydrolysis, releasing the nascent peptide from the ribosome (*Figure 5C*). Dissociation of ArfA and RF2, possibly through the inactive Structure I state (*Figure 5D*), results in ribosomes with a vacant A site and deacylated tRNA in the P site, enabling subsequent recycling of the ribosome.

After submission of this manuscript, several other structures of 70S•ArfA•RF2 complexes with RF2 in the extended conformation were reported (*James et al., 2016*; *Huter et al., 2017*; *Ma et al., 2017*; *Zeng et al., 2017*). Structure I, however, was not found in these studies. A compact conformation of RF2, similar to that in Structure I, was observed in the presence of a heterologous ArfA•RF2 system or the ArfA A18T mutant (*James et al., 2016*). The observation of both the compact and extended RF2 conformations in a single dataset in our work could be the result of different conditions of complex preparation and/or data processing and classification (see Materials and methods). In summary, the reported structures closely agree with our findings and are consistent with the proposed mechanism of ribosome rescue.

The mechanism of ArfA-mediated ribosome rescue is remarkably different from canonical

**Ribosome rescue by ArfA and RF2**

**Video 1.** An animation showing structural transitions during ArfA-mediated ribosome rescue. Four scenes are shown: (1) A view of the complete 70S complex, as in *Figure 1A and B*. The stalled ribosome with a truncated mRNA and a vacant A site, identified in our cryo-EM sample (*Figure 1—figure supplement 1*) is followed by Structure I and Structure II. (2) The opening of RF2, from Structure I to II, coupled with the movement of the L11 stalk. The transition between Structure I and Structure II was generated using the UCSF Chimera 'Morph Conformations' tool. (3) A close-up view of the decoding center, showing the rearrangements of ribosomal nucleotides, ArfA and RF2, coupled with the opening of domain 3 of RF2. (4) A proposed mechanism of ribosome rescue by ArfA and RF2, schematically shown in *Figure 5*. The animation is also available at http://labs.umassmed.edu/korostelevlab/movarfa.gif

translation termination, wherein RF2 accurately defines the lengths of cellular proteins by direct recognition of stop codons in the A site (*Korostelev et al., 2008*; *Weixlbaumer et al., 2008*). While ArfA converts RF2 into a stop-codon-independent release factor, our structures show that: (i) ArfA does not mimic a stop codon; (ii) the conserved codon-recognition elements of RF2—including the SPF motif—are not required (*Chadani et al., 2012*); and (iii) instead of interacting directly with RF2, the ribosomal decoding center stabilizes ArfA, which in turn stabilizes an active RF2 conformation. Thus, bacteria have evolved an intricate stress-response mechanism in which a small protein with specific affinity to the stalled ribosome re-purposes a release factor. The ArfA-mediated ribosome rescue highlights an impressive ability of living organisms to co-opt existing cellular mechanisms for different and sometimes mutually exclusive purposes.

## Materials and methods

### Preparation of ArfA and RF2

The gene encoding *E. coli* ArfA (ASKA Clone(-) library, National BioResource Project, NIG, Japan) was subcloned into pET24b+ (Novagen) kanamycin resistance vector using the primer set CCCG<u>CA-TATG</u>CATCACCATCACCATCACATGAGTCGATATCAGCATACTAAAGGGC/CCCG<u>GGATCC</u>GTGA TTTACTTTCTTGCCAC containing the NdeI/BamHI restriction sites (underlined) and transformed into an *E. coli* BLR/DE3 strain. The resulting ArfA protein is 60 amino acids long and is N-terminally His$_6$-tagged. DNA sequencing confirmed the native sequence of the ArfA gene. Cells containing the pET24b+ plasmid were cultured in Luria-Bertani (LB) medium with 50 μg mL$^{-1}$ kanamycin at 37°C until the OD$_{600}$ reached 0.7–0.8. Expression of ArfA was induced by 1 mM IPTG (Gold Biotechnology Inc., USA), followed by cell growth for 9 hr at 16°C. The cells were harvested, washed and resuspended in buffer A (50 mM Tris-HCl (pH 7.5), 150 mM KCl, 10 mM imidazole, 6 mM $\beta$-mercaptoethanol ($\beta$ME) and protease inhibitor (complete Mini, EDTA-free protease inhibitor tablets, Sigma Aldrich, USA). The cells were disrupted with a microfluidizer (Microfluidics, USA), and the soluble fraction was collected by centrifugation at 18,000 rpm for 20 min and filtered through a 0.22 μm pore size sterile filter (CELLTREAT Scientific Products, USA).

ArfA was purified in three steps. The purity of the protein after each step was verified by 12% SDS-PAGE stained with Coomassie Brilliant Blue R 250 (Sigma-Aldrich). First, affinity chromatography with Ni-NTA column (Nickel-nitrilotriacetic acid, 5 ml HisTrap, GE Healthcare) was performed using FPLC (Äkta explorer, GE Healthcare). The cytoplasmic fraction was loaded onto the column equilibrated with buffer A and washed with the same buffer. ArfA was eluted with a linear gradient of buffer B (buffer A with 0.5 M imidazole). Fractions containing ArfA were pooled and dialyzed against buffer C (buffer A without imidazole). The protein then was purified by ion-exchange chromatography (5 ml HiTrap FF Q-column, GE Healthcare; FPLC). The column was equilibrated and washed with Buffer C, the protein was loaded in Buffer C and eluted with linear gradient of Buffer D (Buffer C with 1 M KCl). Finally, the protein was dialyzed against 50 mM Tris-HCl buffer (pH 7.5), 150 mM KCl, 6 mM $\beta$ME and protease inhibitor (complete Mini, EDTA-free protease inhibitor tablets, Sigma Aldrich, USA) and purified using size-exclusion chromatography (Hiload 16/60 Superdex 75 pg column, GE Healthcare). The fractions of the protein were pulled, buffer exchanged (25 mM Tris-HCl (pH 7.0), 50 mM K(CH$_3$COO), 10 mM Mg(CH$_3$COO)$_2$, 10 mM NH$_4$(CH$_3$COO) and 6 mM $\beta$ME) and concentrated with an ultrafiltration unit using a 3 kDa cutoff membrane (Millipore). The concentrated protein was flash-frozen in liquid nitrogen and stored at −80°C.

N-terminally His$_6$-tagged RF2 (*E. coli* K12 strain) was purified as described (*Korostelev et al., 2008*; *Laurberg et al., 2008*).

### Preparation of the 70S rescue complex bound with ArfA•RF2

70S ribosomes were prepared from *E. coli* (MRE600) as described (*Moazed and Noller, 1986*, *1989*), and stored in the ribosome-storage buffer (20 mM Tris-HCl (pH 7.0), 100 mM NH$_4$Cl, 12.5 mM MgCl$_2$, 0.5 mM EDTA, 6 mM $\beta$ME) at −80°C. Ribosomal 30S and 50S subunits were purified using sucrose gradient (10–35%) in a ribosome-dissociation buffer (20 mM Tris-HCl (pH 7.0), 300 mM NH$_4$Cl, 1.5 mM MgCl$_2$, 0.5 mM EDTA, 6 mM $\beta$ME). The fractions containing 30S and 50S subunits were collected separately, concentrated and stored in the ribosome-storage buffer at −80°C. *E. coli* tRNA$^{fMet}$ was purchased from Chemical Block. RNA, containing the Shine-Dalgarno sequence and a

linker to place the AUG codon in the P site (GGC AAG GAG GUA AAA <u>AUG</u>) was synthesized by IDT.

The 70S•mRNA•tRNA$^{fMet}$•ArfA•RF2 complex was prepared by reconstitution in vitro. 2 µM 30S subunit (all concentrations are specified for the final solution) were pre-activated at 42°C for 5 min in the ribosome-reconstitution buffer (20 mM Tris-HCl (pH 7.0), 100 mM NH$_4$Cl, 20 mM MgCl$_2$, 0.5 mM EDTA, 6 mM $\beta$ME). After pre-activation, 1.8 µM 50S subunit with 24 µM mRNA and 12 µM tRNA$^{fMet}$ were added to the 30S solution and incubated for 15 min at 37°C. ArfA and RF2 were then added at 16 µM each and the solution was incubated for 15 min at 37°C and cooled down to room temperature. The solution was aliquoted, flash-frozen in liquid nitrogen and stored at −80°C.

## Activity of ArfA and RF2

Activity of ArfA and RF2 in the ArfA-mediated rescue was tested using [$^{35}$S]-formylmethionine release assay, essentially as we described previously (*Svidritskiy et al., 2013*). The pre-termination complex was formed using *E. coli* 70S ribosomes, [$^{35}$S]-labeled fMet-tRNA$^{fMet}$ ($^{35}$S-labeled methionine from Perkin Elmer) and truncated mRNA described above. Consistent with published data (*Chadani et al., 2012*; *Shimizu, 2012*), neither ArfA nor RF2 alone induced release of [S$^{35}$]-fMet from the pre-termination complex. Similarly, a combination of ArfA and RF1 did not result in [S$^{35}$]-fMet release, consistent with the requirement for RF2. By contrast, efficient release was observed in 10 min after addition of ArfA and RF2 in combination, consistent with published data (*Zeng and Jin, 2016*). The time of the cryo-EM sample incubation prior to freezing (15 min, as described in the previous section) was therefore sufficient for achieving equilibrium and peptide release, suggesting that the cryo-EM structures represent interconverting equilibrium states.

The release activity was tested as follows. 70S ribosome complex was formed by incubation of 0.7 µM (all concentrations are given for the final complex) of the 30S subunit at 42°C for 5 min, followed by addition of 0.7 µM of the 50S subunit, 3.5 µM truncated mRNA and incubation at 37°C for 20 min (20 mM Tris acetate (pH 6.5), 100 mM ammonium acetate and 20 mM magnesium acetate). The mixture was cooled to room temperature. 1 µM RF2, 10 µM ArfA and 150 nM [$^{35}$S]-fMet-tRNA$^{fMet}$ were added to the solution to form the rescue complex (ArfA or RF2 were also tested separately in independent experiments). 10 µl of the rescue complex were immediately quenched in 0.1 M HCl to represent a zero-time point. 10 µl aliquots collected after 10 min and 1 hr were quenched in 30 µl of 0.1 M HCl. [$^{35}$S]-fMet-tRNA$^{fMet}$ was extracted with 700 µl of ethylacetate; 600 µl of the extract were mixed with 3.5 ml of a scintillation cocktail (Econo-Safe). Samples were quantified using a scintillation counter (Beckman).

## Cryo-EM and image processing

Holey-carbon grids (C-flat 2/2) were exposed to a 75% argon/25% oxygen plasma for 20 s using a Solarus 950 plasma cleaning system. The forward RF target was set to 7w. Before being applied to the grids, the 70S•mRNA•tRNA$^{fMet}$•ArfA•RF2 complex was diluted in the ribosome-reconstitution buffer supplemented with ArfA and RF2 to the following final concentrations: ~0.45 µM 70S, 6 µM mRNA, 3 µM tRNA$^{fMet}$, 10 µM ArfA and 10 µM RF2. 2 µl of the 70S•mRNA•tRNA$^{fMet}$•ArfA•RF2 complex was applied to the grids. The grids were blotted for 5 s at blotting power 8 at 4°C and ~95% humidity and plunged into liquid ethane using an FEI Vitrobot MK4. The grids were stored in liquid nitrogen.

A dataset of 539,311 particles was collected as follows. 3760 movies were collected using Leginon (*Suloway et al., 2005*) on an FEI Krios microscope operating at 300 kV equipped with a DE-20 Camera System (Direct Electron, LP, San Diego, CA) with −0.5 to −3.0 µm defocus. Each exposure was acquired with continuous frame streaming at 32 frames per second (fps) with various exposure lengths (38, 40, 54, 57 and 72 frames per movie) yielding a total dose of 61 e$^-$/Å$^2$. The nominal magnification was 29,000 and the calibrated pixel size at the specimen level was 1.215 Å. The frames for each movie were processed using DE_process_frames script (in EMAN2 [*Tang et al., 2007*]) which is available from Direct Electron at http://www.directelectron.com/scripts. The movies were motion-corrected and frame averages were calculated using the first half of each movie (data up to a dose of ~30 e$^-$/Å$^2$) and excluding the first two frames, after multiplication with the corresponding gain reference. CTFFIND4 (*Rohou and Grigorieff, 2015*) was used to determine defocus values for each resulting frame average. 503 movies with large drift, low signal, heavy ice contamination, or very thin

ice were excluded from further analysis after inspection of the averages and the power spectra computed by CTFFIND4. Particles were semi-automatically picked from full-sized images in EMAN2 using ~50 particles picked manually to serve as a reference. 320 × 320 pixel boxes with particles were extracted from images and normalized. The stack and FREALIGN parameter file were assembled in EMAN2. To speed up data processing, a 4x-binned image stack was prepared using EMAN2.

Data classification is summarized in *Figure 1—figure supplement 1*. FREALIGN v9 (versions 9.10–9.11) was used for all steps of refinement and reconstruction (*Grigorieff, 2016*). The 4x-binned image stack (539,311 particles) was initially aligned to a ribosome reference (PDB 4V4A) (*Vila-Sanjurjo et al., 2003*) using 3 cycles of mode 4 (search and extend) alignment including data in the resolution range from 300 Å to 30 Å until the convergence of the average score. Subsequently, the 4x binned stack was aligned against the common reference resulting from the previous step, using mode 1 (refine) in the resolution ranges 300–18 Å and 300–12 Å (for both ranges, 3 cycles of mode one were run). In the following steps, the 4x binned stack was replaced by the unbinned (full-resolution) image stack, which was successively aligned against the common reference using mode 1 (refine), including gradually increasing resolution limits (increments of 1 Å, five cycles per each resolution limit) up to 6 Å. The resolution of the resulting common reference was 3.29 Å (Fourier Shell Correlation (FSC) = 0.143). Subsequently, the refined parameters were used for classification of the unbinned stack into 10 classes in 30 cycles using the resolution range of 300–6 Å. This classification revealed seven high-resolution classes and three low-resolution (junk) classes (*Figure 1—figure supplement 1*). The particles assigned to the high-resolution classes that contained RF2 and ArfA were extracted from the unbinned stack (with >50% occupancy and scores >0) using merge_classes.exe (part of the FREALIGN distribution), resulting in a stack containing 320,895 particles. Classification of this stack was performed for 30 cycles using a focused spherical mask around the A site (55 Å radius, as implemented in FREALIGN). This classification yielded three high-resolution classes, two of which contained both ArfA and RF2 (Structures I and II) and one with a vacant mRNA tunnel and A site. Using more classes (up to 8) did not yield additional structures (e.g. containing ArfA alone, RF2 alone or additional ArfA•RF2 conformations). For the classes of interest (Structures I and II), particles with >50% occupancy and scores >0 were extracted from the unbinned stack. Refinement to 6 Å resolution using mode 1 (five cycles) resulted in ~3.15 Å maps (FSC = 0.143). The maps were sharpened using automatically calculated B-factors (approximately $-90$ Å$^2$) in bfactor.exe (part of the FREALIGN distribution) and used for model building and structure refinements. B-factors of $-120$ or $-150$ Å$^2$ were also used to interpret high-resolution details in the ribosome core regions. FSC curves were calculated by FREALIGN for even and odd particle half-sets.

Following the submission of our manuscript, several groups reported 70S•ArfA•RF2 cryo-EM structures (*James et al., 2016*; *Huter et al., 2017*; *Ma et al., 2017*; *Zeng et al., 2017*), but each dataset contained a single ArfA•RF2 conformation (similar to our Structure I and Structure II), unlike our dataset, which contained both ArfA•RF2 states. We hypothesize that this difference may be due to several factors. First, different dataset sizes and processing strategies may have influenced the identification of particle classes. Most notably, we use FREALIGN in all our work for particle alignment and classification, consistently yielding multiple states (*Svidritskiy et al., 2014*; *Koh et al., 2014*; *Abeyrathne et al., 2016*; *Loveland et al., 2016*). In some cases, this contrasted the results from other groups that reported only single states. Most recently, we reported several globally different 70S•RelA states from a single sample (*Loveland et al., 2016*) while work on similar (but not identical) 70S•RelA complexes using different data processing approaches yielded only a single state (*Arenz et al., 2016*; *Brown et al., 2016*). However, clarification of the role of FREALIGN in our consistent discovery of multiple structures requires further testing, including reprocessing of datasets from different groups with different software. Second, different buffer conditions, strategies of complex formation (e.g. the use of individual ribosomal subunits in our work) and constructs (e.g. ribosome, RF2 or ArfA strains) may have led to different equilibria between Structure I and Structure II. The presence of the His$_6$-tag on ArfA in our work may have led to stabilization of Structure I. In the absence of density for His$_6$ in our maps, however, we cannot explain how it may have stabilized the structure.

## Model building and refinement

Recently reported cryo-EM structure of *E. coli* 70S•RelA•A/R-tRNA[Phe] complex (*Loveland et al., 2016*), excluding RelA and tRNA[Phe], was used as a starting model for structure refinement. The structure of compact RF2 (Structure I) was built using the crystal structure of free RF2 (PDB 1GQE) (*Vestergaard et al., 2001*) as a starting model. The extended form of RF2 (Structure II) was created by homology modeling from *Thermus thermophilus* RF2 within a 70S termination complex (*Korostelev et al., 2008*) using SWISS-PROT (*Bairoch et al., 2004*). ArfA was modeled de novo in Coot (RRID:SCR_014222) (*Emsley and Cowtan, 2004*), using an initial structure predicted by ROBETTA (*Kim et al., 2004*). The secondary structure in our resulting model of AfrA is consistent with those predicted by ROBETTA and I-TASSER (RRID:SCR_014627) (*Yang et al., 2015*). Initial protein and ribosome domain fitting into cryo-EM maps was performed using Chimera (RRID:SCR_004097) (*Pettersen et al., 2004*), followed by manual modeling using Pymol (RRID:SCR_000305) (*DeLano, 2002*) and Coot. The linkers between the domains and parts of the domains that were not well defined in the cryo-EM maps (e.g. loops of RF2 in Structure I, shown in *Figure 1—figure supplement 4A*) were modeled as protein or RNA backbone.

Structures I and II were refined by real-space simulated-annealing refinement using atomic electron scattering factors (*Gonen et al., 2005*) in RSRef (*Chapman, 1995*; *Korostelev et al., 2002*) as described (*Svidritskiy et al., 2014*). Secondary-structure restraints, comprising hydrogen-bonding restraints for ribosomal proteins and base-pairing restraints for RNA molecules, were employed as described (*Korostelev et al., 2008*). Refinement parameters, such as the relative weighting of stereochemical restraints and experimental energy term, were optimized to produce the stereochemically optimal models that closely agree with the corresponding maps. In the final stage, the structures were refined using phenix.real_space_refine (RRID:SCR_014224) (*Adams et al., 2010*), followed by a round of refinement in RSRef applying harmonic restraints to preserve protein backbone geometry. Ions were modeled as $Mg^{2+}$, filling the difference-map peaks using CNS (*Brunger, 2007*). To this end, the maps were converted to structure factors using phenix.map_to_structure_factors (*Adams et al., 2010*). The refined structural models closely agree with the corresponding maps, as indicated by high correlation coefficients of >0.8 and low real-space R-factors of 0.19 for Structures I and II. The resulting models have excellent stereochemical parameters, characterized by low deviation from ideal bond lengths and angles, low number of protein-backbone outliers (no outliers in ArfA) and other robust structure-quality statistics, as shown in *Table 1*. Structure quality was validated using MolProbity (RRID:SCR_014226) (*Chen et al., 2010*).

Structure superpositions and distance calculations were performed in Pymol. The cryo-EM maps for Structure I and II were deposited in the EMDB (EMD-8522 and EMD-8521, respectively) (RRID: SCR_006506). PDB coordinates for Structures I and II were deposited in the RCSB (PDB codes 5U9G and 5U9F, respectively) (RRID:SCR_012820). Figures were prepared in Pymol and Chimera (*DeLano, 2002*; *Pettersen et al., 2004*).

## Sequence and structural analysis

NCBI (PSI) BLAST (RRID:SCR_004870) (*Altschul et al., 1997*) was used to obtain ~400 non-redundant ArfA homolog sequences with less than 95% identity to that of *E. coli* ArfA. MUSCLE (RRID:SCR_011812) (*Edgar, 2004*) was used to generate a multiple-sequence alignment which was presented with WebLogo 3 (RRID:SCR_010236) (*Crooks et al., 2004*) (*Figure 2*).

## Acknowledgements

We thank the National Institute of Genetics, Shizouka, Japan for providing ArfA- and RF2-overexpressing strains of *E. coli* (ASKA- library); Yevheniya Stepanyuk for assistance with ArfA subcloning; Michael Spilman (Direct Electron, LP) for assistance with data collection and processing; Anna B Loveland for assistance with data processing; Darryl Conte Jr for assistance with manuscript preparation; members of the Korostelev laboratory for helpful discussions and comments on the manuscript. This study was supported by NIH Grants R01 GM106105 and GM107465 (to AAK). All authors contributed to manuscript finalization in the acknowledgements section

## Additional information

### Competing interests

NG: Reviewing editor, *eLife*. The other authors declare that no competing interests exist.

### Funding

| Funder | Grant reference number | Author |
| --- | --- | --- |
| National Institutes of Health | GM106105 | Andrei A Korostelev |
| National Institutes of Health | GM107465 | Andrei A Korostelev |

The funders had no role in study design, data collection and interpretation, or the decision to submit the work for publication.

### Author contributions

GD, Conceptualization, Data curation, Software, Formal analysis, Validation, Investigation, Visualization, Methodology, Writing—original draft, Project administration, Writing—review and editing, Conceived and designed the project, Prepared the ribosome complex and processed cryo-EM data, Built structural models and wrote the manuscript, Tested ArfA and RF2 in peptide release assays, Contributed to manuscript finalization; ES, Formal analysis, Assisted with protein and ribosome purification, Tested ArfA and RF2 in peptide release assays, Contributed to manuscript finalization; RM, Formal analysis, Assisted with protein and ribosome purification, Contributed to manuscript finalization; RD-A, TG, NG, Data curation, Software, Formal analysis, Methodology, Assisted with cryo-EM data collection and processing of preliminary datasets, Contributed to manuscript finalization; DS, Data curation, Software, Formal analysis, Methodology, Assisted with cryo-EM sample preparation, Collected cryo-EM data and assisted with initial data processing, Contributed to manuscript finalization; AAK, Resources, Data curation, Software, Formal analysis, Supervision, Funding acquisition, Validation, Investigation, Visualization, Methodology, Writing—original draft, Project administration, Writing—review and editing, Conceived and designed the project, Built structural models and wrote the manuscript, Collection and processing of preliminary datasets, Contributed to manuscript finalization

### Author ORCIDs

Gabriel Demo, http://orcid.org/0000-0002-5472-9249
Nikolaus Grigorieff, http://orcid.org/0000-0002-1506-909X
Andrei A Korostelev, http://orcid.org/0000-0003-1588-717X

## Additional files

### Major datasets

The following datasets were generated:

| Author(s) | Year | Dataset title | Dataset URL | Database, license, and accessibility information |
| --- | --- | --- | --- | --- |
| Demo G, Svidritskiy E, Madireddy R, Diaz-Avalos R, Grant T, Grigorieff N, Sousa D, Korostelev AA | 2017 | 3.2 A cryo-EM ArfA-RF2 ribosome rescue complex (Structure I) | http://www.rcsb.org/pdb/explore/explore.do?structureId=5U9G | Publicly available at the RCSB Protein Data Bank (accession no: 5U9G) |
| Demo G, Svidritskiy E, Madireddy R, Diaz-Avalos R, Grant T, Grigorieff N, Sousa D, Korostelev AA | 2017 | 3.2 A cryo-EM ArfA-RF2 ribosome rescue complex (Structure I) | http://emsearch.rutgers.edu/atlas/8522_summary.html | Publicly available at EMDataBank (accession no. EMD-8522) |
| Demo G, Svidritskiy E, Madireddy R, | 2017 | 3.2 A cryo-EM ArfA-RF2 ribosome rescue complex (Structure II) | http://www.rcsb.org/pdb/explore/explore.do? | Publicly available at the RCSB Protein |

| | | | | |
|---|---|---|---|---|
| Diaz-Avalos R, Grant T, Grigorieff N, Sousa D, Korostelev AA | | | structureId=5U9F | Data Bank (accession no: 5U9F). |
| Demo G, Svidritskiy E, Madireddy R, Diaz-Avalos R, Grant T, Grigorieff N, Sousa D, Korostelev AA | 2017 | 3.2 A cryo-EM ArfA-RF2 ribosome rescue complex (Structure II) | http://emsearch.rutgers.edu/atlas/8521_summary.html | Publicly available at EMDataBank (accession no. EMD-8521). |

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
