## [Decision Letter]

Thank you for submitting your article "Mechanism of ribosome rescue by ArfA and RF2" for consideration by *eLife*. Your article has been favorably evaluated by John Kuriyan (Senior Editor) and three reviewers, one of whom, Rachel Green (Reviewer #1), is a member of our Board of Reviewing Editors. The following individual involved in review of your submission has agreed to reveal their identity: Venki Ramakrishnan (Reviewer #2).

The reviewers have discussed the reviews with one another and the Reviewing Editor has drafted this decision to help you prepare a revised submission.

We have received comments from the reviewers and all three support the publication of this manuscript in *eLife* pending revisions. The reviewers all appreciate that there is new information on ArfA and RF2 binding to the ribosome, even in light of recent publications on this same topic from multiple groups. However, this study will need to be better placed within that (new) context, in particular addressing the key question as to why the alternative compact conformation was so readily observed in this sample, even though a wild type ArfA protein was utilized. We look forward to receiving a revised submission addressing these points raised by the reviewers.

*Reviewer #1:*

This manuscript describes two structures of ArfA and RF2 bound to bacterial ribosomes, in two distinct states, one likely active to promote peptide and the other in a state either that precedes or follows peptide release. The manuscript clearly describes distinct features of these structures relative to previously described structures of related systems (RF2 on stop codon programmed ribosomes and to a lesser extent SmpB and ArfB bound ribosomes) and attempts to reconcile the structures with previous mutational analyses. Despite the recent publications from competitors in Nature and Science, this manuscript still has merit in particular because it presents these two different structures that (almost certainly) reveal something about relevant intermediates in this rescue pathway. Of course, these papers now published will need to be cited and comparisons to them made in a revised manuscript. Overall, I find the manuscript suitable for publication and hope that the authors will address several specific points below:

1) References to ArfA in "pathogenic bacteria" seems gratuitous and not informative – I would eliminate this (Introduction, first paragraph).

2) Figure 1 – panels A and B – ArfA not red enough to look different in color from orange P site tRNA – distinction better in subsequent panels.

3) The discussion of commonalities with ArfB and SmpB could be extended and some of the biochemistry on this topic alluded to (the role for an extended α helix on SmpB, for example, could be discussed).

4) At the risk of being self-serving, work from my group from 2010 (https://www.ncbi.nlm.nih.gov/pubmed/20208546) documented conformational changes in the positioning of RF1 on ribosomes as a result of cognate (relative to near-cognate) stop codon recognition (using tethered BABE probing). I wonder if it might be interesting to compare some of these structural events by admittedly a lower resolution study to the current "induced" state seen here by cryoEM.

*Reviewer #2:*

This is an important paper that describes an alternate rescue system in bacteria for ribosomes that are stalled on an mRNA without a stop codon. It involves a protein, ArfA, that binds to the A site of the ribosome in the absence of message in the A site, and recruits release factor RF2. Binding induces a conformational change that results in the active form of RF2 that cleaves the nascent peptide. The authors have used cryoEM to capture both the initial inactive state as well as the final activated state from the same sample.

1) Since this paper was submitted, three other structures of similar complexes have been published. Therefore, the paper could be improved by analyzing it in the light of these other results without really detracting from the authors' credit (their BioRxiv preprint clearly establishes their work was independent and nearly simultaneous). For example, the compact inactive RF2 conformation is no longer "hitherto unobserved" but has now also been reported by James et al. in Science.

2) The two observed states determined from a single sample are puzzling. The recent structures of wild-type ArfA with wild-type RF2 from Huter et al., James et al. and Ma et al. only report a single conformation. It seems unlikely that the difference is due to different EM processing schemes as the compact class in this manuscript is not a minor species, but is even more prevalent than the extended class.

James et al. were able to observe the compact conformation, but only by using an ArfA mutant or a heterologous release factor from *Thermus thermophilus*. Mutations in the N-terminus of ArfA or the switch loop of RF2 could potentially cause the observed compact conformation and it may be that the authors have a mixture of wild-type and mutant ArfA.

If there is no mutant protein in the sample, why do the authors think they observe both conformations? Are there substantial differences in the way in which the complex is formed compared to Huter et al., James et al. and Ma et al? Could the uncleaved histidine-tag at the N-terminus of ArfA interfere with complex formation?

3) The authors propose a mechanism and video (Figure 5) from their observed states in which Structure I binds prior to a transition to Structure II. In my opinion the authors are probably correct, but there is some evidence that the release factor exists in two states in solution as shown by one SAXS study (Zoldak et al. 2007. NAR) so the binding of both forms could be independent. Alternatively, the compact conformation may occur after peptide-bond cleavage. Do the authors observe density for the fMet group in both conformations to confirm that both conformations occur prior to peptide bond cleavage? This may be worth discussing.

*Reviewer #3:*

This manuscript describes two cryo-EM structures of ArfA- and RF2-bound ribosomes stalled on a truncated mRNA at 3.2-Å resolution. The two structures, isolated from the same sample, reveal two distinct conformations of RF2, providing information on the structural dynamics of ribosome-bound RF2 during ArfA-mediated ribosome rescue. The work described here clearly advances our understanding of the ArfA-mediated ribosome rescue pathway and we therefore recommend publication of this manuscript in *eLife*.

---

## [Author Response]

*[…] Reviewer #1:*

*[…] 1) References to ArfA in "pathogenic bacteria" seems gratuitous and not informative – I would eliminate this (Introduction, first paragraph).*

We eliminated this sentence.

*2) Figure 1 – panels A and B – ArfA not red enough to look different in color from orange P site tRNA – distinction better in subsequent panels.*

We improved panels A and B in Figure 1 to make ArfA red and distinct from orange of the P-site tRNA.

*3) The discussion of commonalities with ArfB and SmpB could be extended and some of the biochemistry on this topic alluded to (the role for an extended α helix on SmpB, for example, could be discussed).*

We have extended the Discussion and provide a paragraph on commonalities and differences among ArfA, ArfB and SmpB, including the role of an extended α-helix.

*4) At the risk of being self-serving, work from my group from 2010 (https://www.ncbi.nlm.nih.gov/pubmed/20208546) documented conformational changes in the positioning of RF1 on ribosomes as a result of cognate (relative to near-cognate) stop codon recognition (using tethered BABE probing). I wonder if it might be interesting to compare some of these structural events by admittedly a lower resolution study to the current "induced" state seen here by cryoEM.*

We thank the reviewer for pointing this out. We now draw a parallel with stop-codon decoding and provide references to the BABE-probing work by the Green lab and fluorescence-based work by the Simpson lab, which showed conformational rearrangements in release factors.

*Reviewer #2:*

*[…] 1) Since this paper was submitted, three other structures of similar complexes have been published.Therefore, the paper could be improved by analyzing it in the light of these other results without really detracting from the authors' credit (their BioRxiv preprint clearly establishes their work was independent and nearly simultaneous). For example, the compact inactive RF2 conformation is no longer "hitherto unobserved" but has now also been reported by James et al. in Science.*

We have included a discussion of the papers published since the submission of our manuscript.

We have eliminated the expression “hitherto unobserved”.

*2) The two observed states determined from a single sample are puzzling. The recent structures of wild-type ArfA with wild-type RF2 from Huter et al., James et al. and Ma et al. only report a single conformation. It seems unlikely that the difference is due to different EM processing schemes as the compact class in this manuscript is not a minor species, but is even more prevalent than the extended class.*

*James et al. were able to observe the compact conformation, but only by using an ArfA mutant or a heterologous release factor from Thermus thermophilus. Mutations in the N-terminus of ArfA or the switch loop of RF2 could potentially cause the observed compact conformation and it may be that the authors have a mixture of wild-type and mutant ArfA.*

*If there is no mutant protein in the sample, why do the authors think they observe both conformations? Are there substantial differences in the way in which the complex is formed compared to Huter et al., James et al. and Ma et al? Could the uncleaved histidine-tag at the N-terminus of ArfA interfere with complex formation?*

We discuss several factors that may have contributed to the difference of our work from the other reports. Our observation of both conformations is not due to an ArfA mutant, as we state in Methods that “DNA sequencing confirmed the native sequence of the ArfA gene”. Furthermore, we confirmed the enzymatic activity of ArfA (see Methods). Among the likely factors are: (1) Different data processing in our work employing Frealign. This approach also resulted in more than one conformation in our recently published studies of RelA (Loveland et al., 2016) and IRES mRNA (Abeyrathne et al., 2016), unlike the concurrent reports of similar complexes from other labs that used Relion or other strategies and reported a single state in each report (Fernandes et al., *eLife*, 2016; Brown et al., 2016; Arenz et al., 2016). (2) Use of the His6-tag on ArfA. This may have contributed to stabilization of the less abundant Structure I, however we do not observe the His-tag and its potential interactions in cryo-EM maps. We note that Huter et al. had an extension on ArfA resulting from protease cleavage, which did not result in observation of Structure I in their work. (3) Different buffer composition and 70S complex assembly. We assembled the complexes using individual ribosome subunits (purified from endogenous tRNAs and mRNA). Also, buffer conditions, with different concentrations of Mg or other ions could have an effect on the equilibrium between states I and II.

*3) The authors propose a mechanism and video (Figure 5) from their observed states in which Structure I binds prior to a transition to Structure II. In my opinion the authors are probably correct, but there is some evidence that the release factor exists in two states in solution as shown by one SAXS study (Zoldak et al. 2007. NAR) so the binding of both forms could be independent. Alternatively, the compact conformation may occur after peptide-bond cleavage. Do the authors observe density for the fMet group in both conformations to confirm that both conformations occur prior to peptide bond cleavage? This may be worth discussing.*

There is indeed a discussion in the field that RFs exist in solution in both the closed and open forms. Recent studies, however, converge to a view that RFs are predominantly compact (Zoldak et al., 2007; Trappl and Joseph, 2016) and that the initial SAXS report may have overestimated the degree of RF1 opening. We have extended our Discussion to address this point of the reviewer.

As for fMet, we used deacylated tRNA^fMet^ as we state in Methods, and we do not observe density for fMet in the PTC, as expected.